# Government Attention, Market Competition and Firm Digital Transformation

**Xuejun Jin and Xiao Pan ***

School of Economics, Zhejiang University, Hangzhou 310027, China; cec_jxj@cec.zju.edu.cn
* Correspondence: 11701027@zju.edu.cn

**Abstract:** Clarifying the driving factors of enterprise digital transformation can help us understand the real driving forces of industrial digitization and digital industrialization, improve the implementation of industrial policies, and narrow the digital divide between different regions and firms to facilitate high-quality and sustainable development. Based on 38,891 news items from provincial and municipal governments in China, this paper uses text analysis to depict the government's attention to the digital economy and explore the influencing factors driving digital transformation. In the empirical analysis, government attention to the digital economy positively impacts enterprise digital transformation primarily through fiscal expenditures on science and technology, the digital economy level, the digital financial inclusion level, industrial agglomeration, and firm nature. The positive impact of market competition on enterprise digital transformation is significant for small-scale firms. The insight from this finding is that enterprise digital transformation cannot be solved entirely by market forces but also needs to be led by digital industrial policies with government attention.

**Keywords:** digital transformation; government attention; market competition; text analysis





## 1. Introduction

In recent years, digital transformation has profoundly changed the way enterprises do business, affected the way they establish relationships with consumers, suppliers, and financial institutions, promoted the innovation of enterprise business models, created more value for enterprises, and changed the operational and organizational processes of enterprises as a whole [1]. A new era of the digital economy is now upon us, marked by the development of digital technologies represented by artificial intelligence, big data, and cloud computing. China's economic development has been fueled by digital transformation, which has become one of the major driving forces and a key aspect of competition between regions and companies within the country. However, due to insufficient awareness, limited technical capabilities, high costs, and long transformation cycles, firms do not understand transformation and, thus, do not, cannot, and dare not transform. In terms of quantity, according to the survey data of the China Federation of Industry and Commerce in 2020, the digital transformation of Chinese firms is still in its initial stage, and although more than 50% of firms have already carried out or plan to carry out digital transformation, approximately 37% of firms do not have a transformation plan [2]. In terms of region, there is a regional digital divide regarding digital transformation, and this divide generally weakens from east to west and exhibits a bifurcation trend [3].

From the perspective of top-level design, China's 14th Five-Year Plan indicated acceleration of the development of the digital economy and promotion of the digital transformation of enterprises, and at the same time promulgated the first national special plan in the field of digital economy—the "14th Five-Year Plan" Digital Economy Plan—and remarked that the added value of the core industries of the digital economy should account for 10% of GDP, promoting the development of the digital economy and the digital transformation of enterprises. In the 2022 Chinese Government Work Report, digital economy-related content

occupies the most space, and in recent years, 31 provinces in China have issued relevant plans for the development of the digital economy. For example, Beijing plans to build a benchmark city for the digital economy, accelerate the construction of digital infrastructure, and vigorously develop digital trunking private networks and edge computing systems, as well as 6G technology; Shanghai plans to promote the city's digital transformation; Guangdong plans to establish an industrial Internet demonstration zone; and Henan plans to establish a state-level new Internet exchange center and a national-level data trading venue. To support the construction of the digital economy, the national and local governments have introduced detailed measures for enterprise digitalization projects, key technology research in enterprises, the construction of digital public service platforms, infrastructure construction, and electricity price support. China's national and local governments attach great importance to the digital economy and the digital transformation of enterprises.

From a practical point of view, on the one hand, technological factors promote the digital transformation of enterprises, such as smart devices [4], digital production systems [5,6], software and applications [7], data analysis [8], and infrastructure [9]; on the other hand, intangible drivers also promote the digital transformation of enterprises [10], such as internal factors [11,12], competitive or cooperation factors between enterprises [13–16], and government institutional support and policy drivers [17–19]. Due to China's special economic system, market-oriented reforms are not sufficient, the market economy system is not perfect, resources in many fields have not been effectively allocated, and at the same time, the government often intervenes in the economy. So the government is paying more and more attention to the relationship between the government and the market and hopes that the market and the government can coordinate with each other to jointly promote economic development [20]. In addition, external support and external pressure are the two most important environmental determinants of digital transformation [21]. Therefore, this paper incorporates the market and the government into the analysis framework and studies whether enterprises compete, imitate each other, and influence the market, thereby promoting the digital transformation of enterprises, or whether the government's support and related policies promote the digital transformation of enterprises. Clarifying the driving factors of enterprise digital transformation can help us understand the real driving force of industrial digitization and digital industrialization, improve the implementation effect of industrial policies, and narrow the digital divide between different regions and firms to help promote high-quality and sustainable development.

The contribution of this paper differs from that of the current relevant literature in the following respects: First, this paper is the first to evaluate the level of government attention to the digital economy. Using Python crawler technology and text analysis and based on the report data of 38,891 press releases in 31 provinces in China, the frequency of occurrence of keywords related to "digital economy" is statistically calculated, and government attention to the digital economy is depicted.

Second, from the perspective of the government's attention to the digital economy and market competition, the factors driving digital transformation are studied. The existing literature on digital transformation mainly studies the impact of digital transformation on the market; for example, digital transformation affects business models [22,23], supply chains [24,25], servitization [23,26], and customer relationships [27,28]. There is also literature examining the impact of digital transformation on the economy and society, such as the labor market [29,30], environmental sustainability [31,32], energy efficiency [33], and corporate behavior and performance [34–38]. There are studies on the impact of digital government on the economy [39–42]. The current literature on the factors driving digital transformation comes mostly from case studies, questionnaires, and conceptual analysis, with less empirical research through a large amount of data [10]. The literature on government attention has mostly been qualitative and has focused on the perspective of public management or the regulatory effect of pressure at higher and lower levels of government [43]. This paper conducts empirical research and enriches the study of enterprise digital transformation by combining government attention to the digital economy

and market competition in the analysis, providing more direct empirical evidence and more comprehensive research on the real factors driving enterprise digital transformation. Moreover, this work explains that the degree to which competition plays an important role in promoting digital transformation depends on enterprise size.

Third, this work further examines the impact of government attention on enterprise digital transformation from the perspectives of fiscal science and technology expenditures, digital economy level, digital financial inclusion level, industrial agglomerations, and firm nature channels.

The second part of the paper presents the theoretical framework. The third part of the paper presents the research design. The fourth part of the paper presents the empirical results and analysis. The last part of the paper presents the conclusions.

## 2. Theoretical Framework

### 2.1. Theory and Research on Digital Transformation

In the empirical literature, digital transformation is examined primarily from the following three perspectives: First, the impact of digital transformation on enterprises is examined. Digital transformation can reduce information asymmetry and irrational management behavior and improve corporate governance [44]. Digital transformation can reduce costs and increase asset utilization and profitability [34]. Digital transformation can improve corporate operational efficiency [45], organizational resilience [46], and stock liquidity [35], influence debt costs [36], empower supply chain finance [47], and ease financing constraints. Digital transformation can improve sustainable innovation capability [48], green technology innovation capability [37], total factor productivity [49], and firm value [38]. There is a significant peer effect in the process of enterprise digital transformation [50]. In addition, digital transformation can promote the division of labor [51] and the expansion of corporate exports [52]. The high-quality [53] and sustainable development of enterprises [54–56] are inseparable from digital transformation. Second, the external factors affecting enterprise digital transformation are analyzed. Institutional environment theory states that enterprises are affected by the environment, the external environment influences the adoption of new digital technologies [57,58], and the technological spillover effect of the digital economy is conducive to enterprise digital transformation. The government's competitive policy is important and should be dynamically adjusted [59]. Government spending on science and technology and government subsidies [60] can promote enterprise digital transformation by reducing financing constraints, increasing financial stability, and promoting enterprise innovation [61]. Other government actions also have an impact, such as interest rate liberalization reforms that are conducive to digital transformation [62], and the government's goal of high economic growth inhibits enterprise digital transformation [63]. In addition, the uncertainty of the external environment, the intensity of industry competition, and social network embedding promote the cohort effect of enterprise digital transformation [50]. Government support and IT infrastructure can facilitate digital transformation [64]. Third, the internal factors affecting enterprise digital transformation are analyzed. For example, the financialization of firms inhibits digital transformation [65], the level of supply chain financing increases in favor of the degree of digital transformation [66], and digital mergers and acquisitions contribute to enterprise digital transformation [67]. During the COVID-19 pandemic, digital orientation and capabilities significantly contributed to enterprise digital transformation [68]. Digital strategy, IT resources, and highly skilled employee resources are conducive to enterprise digital transformation [69], and the characteristics of enterprise managers also influence digital transformation [70]. Finally, the impact of digital transformation with respect to the government has also been analyzed in the literature. For example, the digital transformation of the government integrates digital technology into governance by the government [71], reduces administrative corruption, improves government efficiency and the environment of enterprises, promotes enterprise innovation [72], and increases the total factor productivity of enterprises [73].

From other studies, it can be seen that the rapid development of information technology is the most important driver of social change and enterprise transformation [74]. Combined with innovation-driven theory, market factors lead enterprises to transform their business models and carry out digital transformation [75]. The impact of new digital technologies, fierce digital competition, and the corresponding digital customer behaviors are external factors that promote enterprise digital transformation, among which the digital resources owned by enterprises and the organizational structure that matches digitalization and digital growth strategy are the core factors of transformation [76]. A company's strategic vision, innovation culture, digital technology reserves, governance capabilities, innovation capabilities, and access to resources are important factors for the success of digital transformation [77,78]. There are also studies on the drivers of digital transformation in various industries, such as the healthcare [79], automotive [80], energy [81], real estate [82], and finance [83] industries. Government actions also affect enterprise digital transformation, and digital government innovation in South Korea has been on the national agenda in recent years. The president's leadership determines the success or failure of innovation, and thus, digital governance policies need to remain sustainable [84]. Due to the lack of capital, digital capabilities, and human resources of small service enterprises, there are technical barriers, and the support of government policies and programs is important for enterprise digital transformation [85]. Both technical and intangible enablers influence digital transformation [10]. The impact of COVID-19 has accelerated the digitalization of businesses [86,87]. There are also some areas of concern for the impact of digital transformation. Digital technologies are used in the circular economy and benefit economic development [88]. Digitalization contributes to environmental sustainability [32], resource protection [33], and energy efficiency, and can reduce negative environmental impacts [31]. Digital technologies can increase enterprise knowledge management, increase productivity, and reduce costs [89]. The digitization of work and HR processes can improve organizational sustainability [90]. Digitalization can also transform business management processes [91], change human resource management [92], and affect the labor market. Digital transformation will promote workforce upgrading, which can lead to unemployment and the polarization of employment opportunities [29,30]. At the same time, digital transformation has spawned new professions, such as in the field of quality engineering [93], as well as new jobs that are flexible in time and space [19,94]. In China, digital transformation has led to the replacement of low-skilled jobs, increasing the demand for higher qualifications [95].

### 2.2. Analysis of the Mechanism of Government Attention to the Digital Economy Affecting Enterprise Digital Transformation

Attention refers to the selective attention paid to a specific aspect of subjective or objective information and ignores other behaviors and cognitive processes of perceiving information; attention is the priority of one thing over others, and the greatest characteristic of attention is preference [96–98]. Jones and Baumgartner were the first to bring attention to the field of political science [99]. Attention is a scarce resource with differentiated distribution in time and space; attention allocation in government behavior is regarded as a way of discourse; attention is an important issue affecting the government's policy design; and government attention emphasizes the effective allocation of attention resources [43]. The "Attention-Driven Policy Choice Model" proposed by Jones et al. (1993) believes that limited attention and attention shift are the basic reasons for policy stability and policy mutation [100]. In the democratic political systems of developed countries, the degree of public participation in policies is high, and policies can be considered the result of multiparty games. However, China's research emphasizes the central position of the central government and governments at all levels, and China has the characteristics of "strong government-weak society," so China's policies are mostly considered to be the result of the transmission of the government's will [101]. Government attention is very important for the influence of its behavior and economic decisions. Generally speaking, after China's

central government issues the outline of the five-year plan for national economic and social development and various national economic policy documents, local governments will pay close attention to the central policy documents and issue corresponding policies and regulations in various localities. When a local government pays attention to a certain aspect, it investigates and studies this aspect, promulgates a series of plans and policies, and accelerates the approval of government-led projects in the field. Resources exist wherever government attention exists. Enterprise digital transformation is an important part of digital economic development.

In general, as the government pays more attention to the digital economy, it will adopt some methods to directly and indirectly support the digital transformation of enterprises. Its direct support is achieved by promulgating policies, regulations, or plans that promote the development of the local digital economy and provide direct support for the digital transformation of businesses in the local market. The government promulgates fiscal and tax policies, including financial subsidies, tax exemptions, low-interest loans, price subsidies, and government procurement and incentives. As part of the digital economy plan, the government also increases fiscal expenditures for science and technology to support the digital economy and establishes investment funds for the digital economy. The indirect support takes effect through the development of infrastructure and the establishment of a support system for the development of the digital economy. The government has established a digital economy infrastructure, digital economy pilot cities, digital economy industrial parks, and digital trading platforms [102]. Moreover, the government cooperates with universities to promote the development of industry, education, and research for the digital economy and to cultivate digital talent for enterprise digital transformation. Furthermore, the government has promulgated rules, such as those pertaining to intellectual property rights, to ensure that businesses are able to transform themselves digitally.

These direct and indirect support methods facilitate the digital transformation of enterprises for the following reasons: The first benefit of these measures is the provision of funds for the development of local digital economies, the alleviation of financing constraints for enterprises, the reduction in uncertainty surrounding R&D funding for digital technology [61], and financial guarantees to facilitate digital transformation. Second, because the government controls the pricing and distribution rights of factors such as land, capital, and labor, its need to develop the digital economy prompts it to communicate with enterprises [71], reduce their rent-seeking behavior and information asymmetry, reduce the costs and risks faced by enterprises using digital technology, and stimulate their technological innovations [73]. Government behavior is a bellwether for enterprises that actively carry out digital transformation to obtain more government support. Third, the government's construction of digital infrastructure and digital industrial parks has promoted the digitalization of the supply chain and saved upstream and downstream transaction costs [103]. Fourth, the government's emphasis on the digital economy has released signals, reduced the information asymmetry of investors, reduced the cost of identifying enterprises for investors, effectively concentrated financial resources on those enterprises undergoing digital transformation, and reduced the financing costs of enterprises [104]. Fifth, the business environment is a key factor affecting the diffusion of innovation [105]. The diffusion and adoption of information technology cannot be achieved without government support [106], and the government provides a good business environment for enterprise digital transformation and promotes the use and dissemination of digital technologies [107]. Therefore, this paper makes the following assumption:

**Hypothesis 1.** *The government's attention to the digital economy has significantly facilitated the digital transformation of enterprises.*

### 2.3. Analysis of the Mechanism of Market Competition Affecting Enterprise Digital Transformation

Market competition is conducive to enterprise digital transformation. First, in terms of the pressure to catch up, market competition leads firms to feel this pressure from competitors, and then firms adopt new technologies to maintain their competitive advantage [108]. Market competition is an external factor of influence on the business environment of firms, affecting their business model, sales strategy, scale expansion or contraction, and ability to operate and use their assets efficiently. The rising level of market competition and increased market uncertainty lead firms to become more motivated to adopt offensive or defensive competitive behaviors to maintain their competitive advantage [51]. When the number of competitors for new technologies increases in the market, firms that quickly absorb and transform pressure use the idea of adopting new technologies to solve their own strategic needs. If firms know that their peers are using digital transformation technologies, then they sense a crisis and can quickly lose their competitive advantage in the industry without the help of new technologies, so competitive pressures drive digital transformation [109–112]. Conversely, if other firms in the same industry no longer use digital transformation technologies, such firms may not take a step forward toward the use of new technologies. The willingness to adopt new technologies in the face of competitive pressures is more pronounced in low-innovation or low-adventure firms compared to other firms [106]. The high-tech industry is changing rapidly, putting companies under pressure to realize and follow the adoption of competitors' digital transformation technologies more quickly. Companies can use such transformation technologies to better understand market visibility, improve operational efficiency, and obtain more accurate data. A strong competitive atmosphere allows partners to influence a firm's new technology adoption, prompting digital transformation through financial incentives or mandatory requirements from partners with a high degree of bargaining power [113]. The associated technological competition and customer pressure lead firms to replicate industry leader behavior [114], and when industry leaders undergo digital transformation, firms in that industry follow suit. To meet the challenges of dynamic environmental changes, firms combine digital technologies with corporate products and services to gain access to consumer preferences and expand their market share [107]. Second, regarding corporate governance, increased market competition can mitigate principal-agent problems, reduce managerial laxity, improve managerial efficiency [115], and encourage companies with weaker governance structures to increase the speed of their adjustments to the optimal capital structure and thus maximize shareholder wealth [116]. An increase in industry competition can improve the degree of information disclosure and the transparency of the information environment [117] and reduce enterprise information asymmetry and management self-serving behavior. The alleviation of management's principal-agent problem leads to the active adoption of new technologies for digital transformation. Finally, in terms of investments and financing, fierce industry competition can promote corporate investments [118], increase R&D and other investments, promote corporate transformation and upgrades, and improve enterprise digital transformation. However, the urgency of transformation varies across enterprises due to their various sizes, and thus, the impact of competition on enterprise digital transformation may differ across enterprises. Due to the structural inertia of large companies, competitive pressures drive less digital transformation [119]. Therefore, the following hypothesis is proposed:

**Hypothesis 2.** *The rise in market competition affects enterprise digital transformation, and the impact varies across different scales.*

### 3. Study Design

#### 3.1. Methods for Data Collection and Sample Selection

This study includes data on China's A-share-listed companies from 2011 to 2020. Data on macroeconomics and firms are gathered mainly from the CSMAR database, and the deposit data for 31 provinces are derived from the WIND database. Data on press releases

are from the INFOBANK database (China Economic News Database), and the raw text of press releases is crawled using Python. A comprehensive database of economic news reports published by nearly 1000 traditional and online media outlets in China is available through INFOBANK. Raw data for this paper are processed in the following manner: companies associated with the financial sector are excluded; ST and PT companies are excluded; and companies that have commenced public trading during the current year are excluded. To eliminate the effect of extreme values on continuous variables, this study weights all continuous variables at the levels of 1% and 99%.

### 3.2. Descriptions and Definitions of Variables

#### 3.2.1. Dependent Variable

In this study, corporate digital transformation (DT) is the dependent variable. This work measures the degree to which a company has adopted digital transformation by the logarithm of the total word frequency of terms related to digital transformation in its annual report [35], by means of a text analysis. This method produces raw data related to DT, which are published in the CSMAR database. Data in the CSMAR database on DT represent the frequency with which artificial intelligence, blockchain technology, cloud computing technology, big data technology, and digital technology are mentioned in each segment of the annual reports of listed companies. To obtain the total word frequency of DT, this study sums up these words and adds them to the total word frequency. Using these data, indicators of DT are derived by logarithmically processing them.

#### 3.2.2. Core Independent Variables

To measure the factors driving enterprise digital transformation, this paper selects government attention to the digital economy and industry competitiveness as core independent variables, which are described below.

The first independent variable is government attention (GA). First, this work uses the text content of the original **press release** crawled down from the China Economic News Library as a data pool for filtering characteristic words and then the text analysis method to count the word frequency using Python; subsequently, the word frequency of the subdivided vocabulary is counted and added. Finally, the sum of the frequency of digital economy-related words used in China's 31 provinces to measure government attention to the digital economy is obtained. Specifically, this work divides the measurement method into the following three steps: The first step is the acquisition of press releases. This paper uses Python to crawl the press releases of the provincial party secretaries, governors, municipal party secretaries, and mayors of 31 provinces in the INFOBANK database and then cleanses them, deletes duplicate and incomplete entries in the original press releases, and finally obtains a total of 38,891 usable press release data points. The second step aims to extract the keywords. Referring to the literature [35], government attention keywords are divided into five categories, as shown in Figure 1. The third step aims to calculate the frequency. We import the content of the dictionary into Python's custom vocabulary list, use the Jieba Chinese word segmentation module to segment sentences in a provincial press release from a certain year, and obtain fragment phrases. Then we read the negative word list, eliminate words with negative prefixes (1517 negative prefixes such as non, do not, no, and none), calculate the frequency of keyword matching in the custom word list in the press release, and add up the frequency obtained by each press release. Our measurement of government attention (GA) to the digital economy is based on the logarithm of the sum of the frequency of keywords related to DT in a province during a certain year.

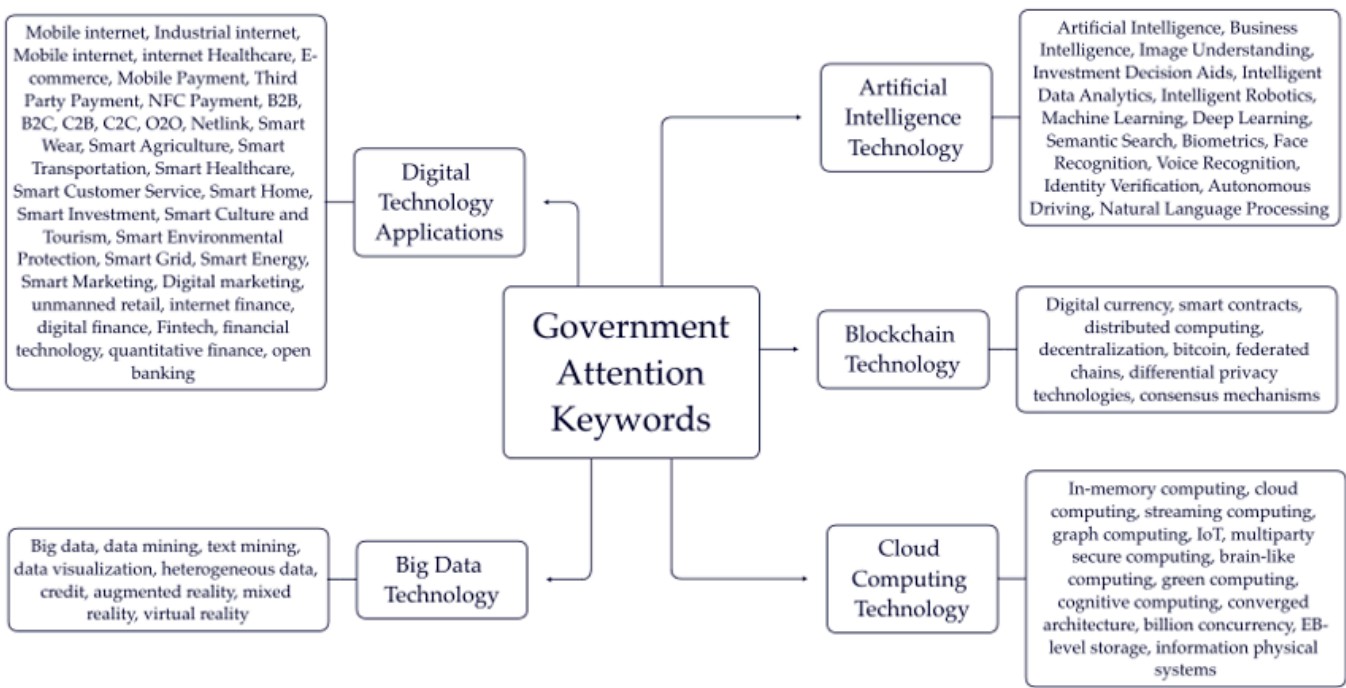

**Figure 1.** Government attention keywords.

The second independent variable is industry competitiveness (HHI). This paper uses the Herfindahl index to measure industry competitiveness [120]. The formula is $HI = \sum_{i}^{N}(X_i/X)^2$, in which $X_i$ represents the total assets of firm i and X represents the sum of the total assets of all firms in the industry to which firm i belongs. A large Herfindahl index indicates a higher degree of monopoly and a smaller degree of competition. To make it more understandable, this paper transforms the Herfindahl data by using $HHI = 1 - HI$; then, an increase in HHI indicates an increase in industry competitiveness.

### 3.2.3. Control Variables

The macro- and firm-level control variables introduced in Equation (1) are regional financial sector development (FD), industry digitization (ID), firm size (Size), leverage (Lev), operating income growth rate (Growth), number of board members (Board), proportion of independent directors (Indep), and duality (Dual). In Table 1, detailed descriptions of the variables are provided.

**Table 1.** Definition of the main variables.

| Variables Names | Symbols | Measurement Method |
|---|---|---|
| Regional financial industry development | FD | Domestic and foreign currency deposit balances by province/GDP by province |
| Industry digitization | ID | Average value of digital transformation for all companies in the industry |
| Size | Size | Natural logarithm of total assets for the year |
| Leverage | Lev | Total liabilities/total assets at the end of the year |
| Operating income growth rate | Growth | Operating income of the current year/operating income of the previous year, minus 1 |
| Number of board members | Board | Natural logarithm of the number of board members |
| Percentage of independent directors | Indep | Number of independent directors/number of directors |
| Duality | Dual | Chairperson and general manager are the same = 1 and 0 otherwise |

### 3.2.4. Descriptive Statistics

Table 2 presents the descriptive statistics of the main variables after the continuous variables have been weighted. Through data processing, we obtained 13,697 sample data points. The results show a mean value of enterprise digital transformation of 1.84, a standard deviation of 1.35, and a maximum value of 5.01. There is a mean value of 3.97, a standard deviation of 1.34, and a maximum value of 5.88 for government attention to the digital economy. There is a mean value of 0.49, a standard deviation of 0.41, and a maximum of 0.96 for industry competitiveness. The median and mean values of digital transformation are relatively close, and digital transformation is common among listed companies. At the same time, all provinces generally attach importance to the digital economy.

**Table 2.** Descriptive statistics of variables.

| Variable Name | Number of Observations | Mean | Standard Deviation | Median | Minimum | Maximum |
| --- | --- | --- | --- | --- | --- | --- |
| DT | 13,697 | 1.84 | 1.35 | 1.79 | 0.00 | 5.01 |
| GA | 13,697 | 3.97 | 1.34 | 4.25 | 0.00 | 5.88 |
| HHI | 13,697 | 0.49 | 0.41 | 0.74 | 0.00 | 0.95 |
| FD | 13,697 | 3.89 | 1.53 | 3.53 | 1.75 | 7.88 |
| ID | 13,697 | 1.82 | 0.79 | 1.62 | 0.30 | 3.82 |
| Size | 13,697 | 22.24 | 1.26 | 22.08 | 19.78 | 26.14 |
| Lev | 13,697 | 0.42 | 0.20 | 0.41 | 0.05 | 0.90 |
| Growth | 13,697 | 0.17 | 0.43 | 0.11 | −0.59 | 2.80 |
| Board | 13,697 | 2.11 | 0.20 | 2.20 | 1.61 | 2.71 |
| Indep | 13,697 | 0.38 | 0.05 | 0.36 | 0.33 | 0.57 |
| Dual | 13,697 | 0.31 | 0.46 | 0.00 | 0.00 | 1.00 |

### 3.2.5. Typical Scenario

In recent years, digital transformation has developed rapidly, and digital technology-related industries have developed steadily. From the perspective of industrial development, digital industrialization and industrial digitalization have developed steadily, and in 2021, the scale of digital industrialization and industrial digitalization would have been 8.4 trillion yuan and 3.72 billion yuan, respectively, as shown in Figure 2.

However, there is an obvious imbalance in regional development in digital transformation, with the eastern region developing better and the western region developing poorly, as shown in Figure 3a. At the same time, the government's attention to the digital economy shows a trend of stronger growth in the east and weaker growth in the west, with the darkest provinces being the most economically developed regions in China. The government's attention to the digital economy reflects obvious regional differences, as shown in Figure 3b.

In accordance with the measurement method described above, this paper presents the mean DT and market competition values of Chinese listed companies as well as the mean values of provincial GA between 2011 and 2020. The results are shown in Figure 4. We find that the overall degree of enterprise digital transformation increased year by year, the degree of market competition remained stable, and the degree of provincial government attention to the digital economy increased significantly in 2013, possibly due to China's Guiding Opinions on Promoting the Orderly and Healthy Development of the Internet of Things and the Special Action Plan for Deep Integration of Informatization and Industrialization (2013–2015) that year. Since then, the government has begun to promote digital technologies such as the Internet of Things; digital technology has gradually entered the industrial field; digital economy policies have begun to germinate; and subsequent relevant policies have been continuously deepened and implemented. It can also be seen from Figure 4 that government attention and enterprise digital transformation have maintained relatively consistent growth.

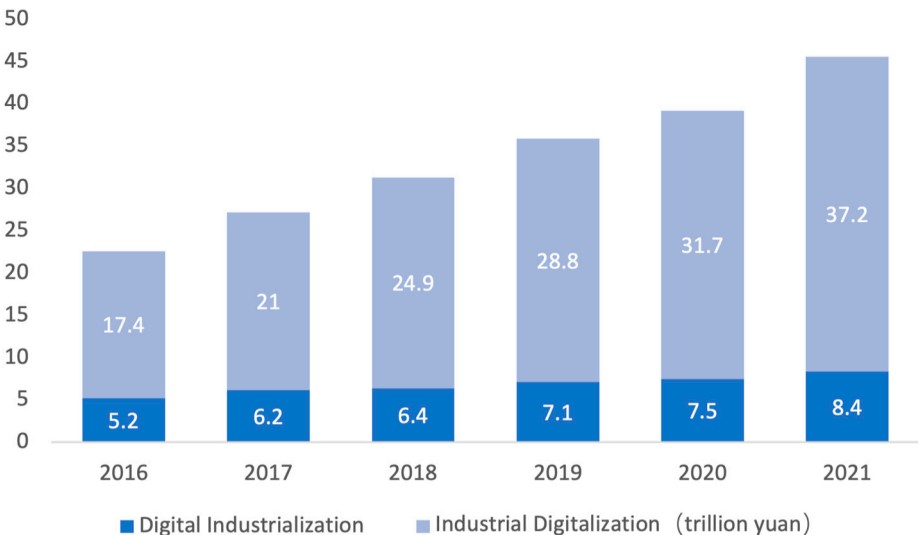

**Figure 2.** The scale of digital industrialization and industrial digitalization. The data is calculated according to the definition of the China Academy of Information and Communications Technology, and the unit is trillion yuan. According to the definition in the report on the development of the digital economy by the China Academy of Information and Communications Technology, digital industrialization refers to the information and communication industry, which provides technical support to the digital economy and is a leading industry; industrial digitalization refers to the penetration of information technology into traditional industries, thereby improving the production efficiency of traditional industries and increasing the output of such industries that belong to the main areas of digital economy development.

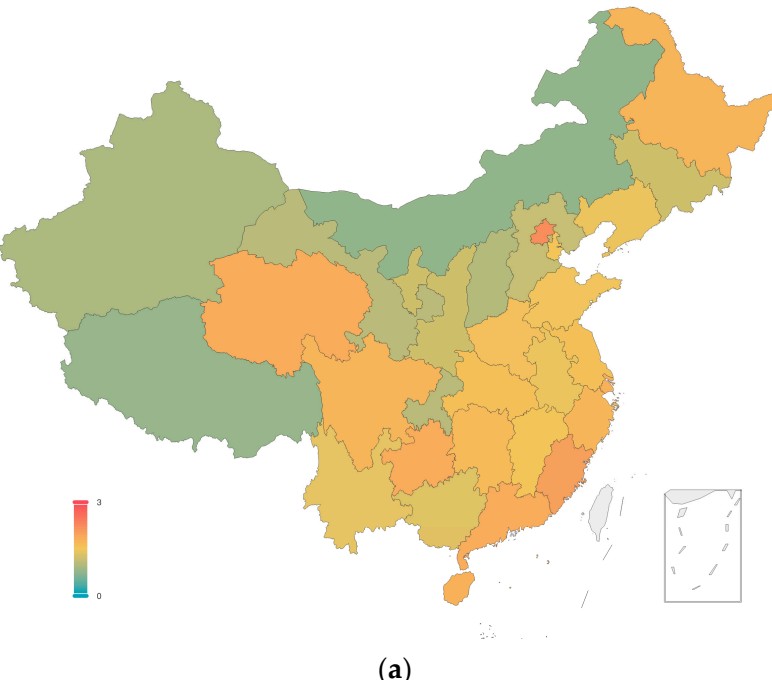

(**a**)

**Figure 3.** *Cont.*

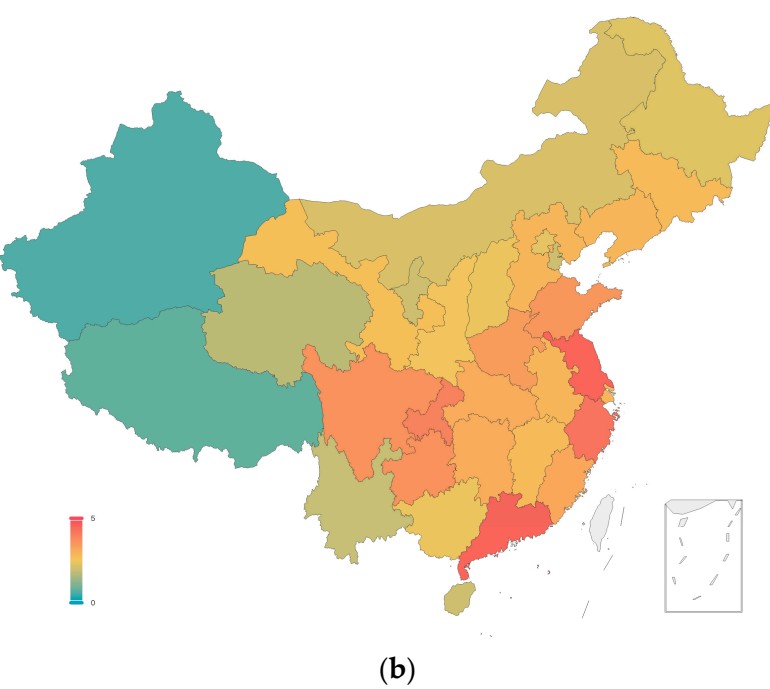

(**b**)

**Figure 3.** Average value of digital transformation and government attention to the digital economy in 31 provinces from 2010 to 2020. The shade of color represents the size of the value, and the darker the color, the larger the value. (**a**) shows the average value of enterprise digital transformation, and the variable definition is shown in Section 3.2.1; (**b**) shows the average government attention to the digital economy, and the variables are defined in Section 3.2.2.

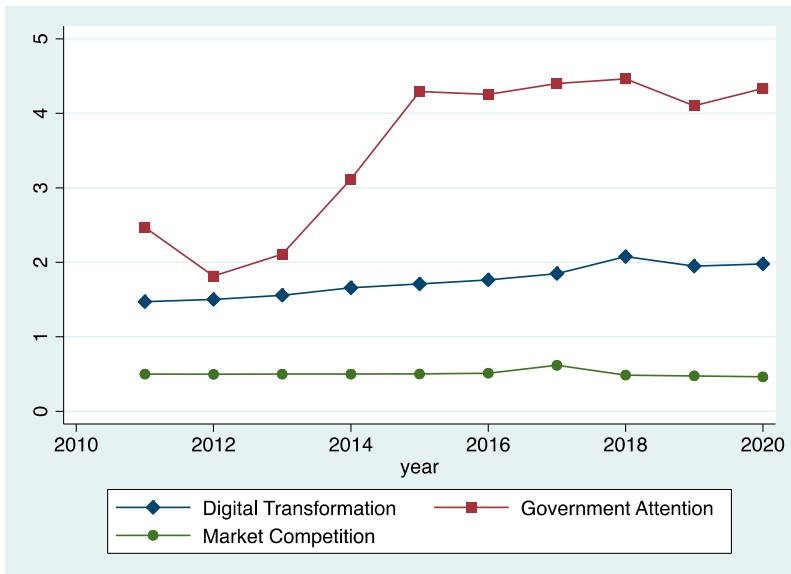

**Figure 4.** Trends in government attention to the digital economy, market competition, and enterprise digital transformation from 2011 to 2020.

### 3.3. Baseline Model Setting

To test the impact of government attention and industry competitiveness on enterprise digital transformation, the following equation is estimated:

$$DT = \beta_0 + \beta_1 GA + \beta_2 HHI + \gamma \sum Controls + \sum FE + \varepsilon, \tag{1}$$

In Equation (1), the dependent variable is enterprise digital transformation (DT), and the independent variables are government attention (GA) and industry competitiveness (HHI). $\beta_1$ portrays the effect of GA on DT, $\beta_2$ portrays the effect of HHI on DT, and Controls denotes the control variables that may affect the dependent variables, as previously mentioned. It is necessary to control time and industry fixed effects to absorb the fixed effects as much as possible, and clustering adjustments are made for standard errors at the industry level.

## 4. Empirical Results and Analysis

### 4.1. Impact of Government Attention and Industry Competitiveness on Enterprise Digital Transformation

As seen in Table 3, the benchmark regression results are presented, showing the results of the estimation without and with the addition of control variables for adjusting for industry and year fixed effects or clustering, respectively, in Columns (1) to (4). As seen, all four regression coefficients of government attention are positive and significant at the 1% level; that is, an increase in government attention to the digital economy significantly promotes enterprise digital transformation. The core independent variable, government attention (GA) to the digital economy, has a coefficient of 0.03, which is significant at the 1% level. The degree of GA significantly promotes DT. Thus, if government attention to the digital economy increases by 1 standard deviation, then enterprise digital transformation increases by 4.11%, which is equivalent to a 2.25% increase in the DT sample mean. Overall, the above finding verifies Hypothesis 1 of this paper. The coefficient of the core independent industry competitiveness (HHI) variable is shown to not be significant in Columns (1) to (4), implying that market competition is not the main factor driving enterprise digital transformation.

**Table 3.** Drivers of enterprise digital transformation.

| | (1) | (2) | (3) | (4) |
|---|---|---|---|---|
| | DT | DT | DT | DT |
| GA | 0.031 *** | 0.031 *** | 0.031 *** | 0.031 *** |
| | (3.46) | (3.38) | (3.90) | (3.58) |
| HHI | −0.010 | 0.050 | −0.010 | 0.050 |
| | (−0.45) | (0.34) | (−0.52) | (1.25) |
| FD | 0.020 *** | 0.022 *** | 0.020 | 0.022 |
| | (2.93) | (3.21) | (0.95) | (1.08) |
| ID | 1.031 *** | 1.030 *** | 1.031 *** | 1.030 *** |
| | (80.06) | (45.87) | (62.61) | (40.04) |
| Size | 0.117 *** | 0.125 *** | 0.117 *** | 0.125 *** |
| | (12.71) | (13.26) | (8.70) | (9.53) |
| Lev | −0.272 *** | −0.231 *** | −0.272 *** | −0.231 *** |
| | (−5.01) | (−4.09) | (−4.45) | (−3.36) |
| Growth | 0.110 *** | 0.109 *** | 0.110 *** | 0.109 *** |
| | (5.01) | (4.94) | (4.85) | (4.97) |
| Board | −0.031 | −0.035 | −0.031 | −0.035 |
| | (−0.52) | (−0.59) | (−0.25) | (−0.28) |
| Indep | 0.030 | 0.017 | 0.030 | 0.017 |
| | (0.14) | (0.08) | (0.07) | (0.04) |
| Dual | 0.107 *** | 0.106 *** | 0.107 *** | 0.106 *** |
| | (5.24) | (5.13) | (3.58) | (3.45) |
| Constant | −2.601 *** | −2.825 *** | −2.601 *** | −2.825 *** |
| | (−10.78) | (−9.91) | (−5.66) | (−5.54) |
| Industry FE | No | Yes | No | Yes |
| Year FE | Yes | Yes | Yes | Yes |
| Observations | 13,697 | 13,697 | 13,697 | 13,697 |
| adj. $R^2$ | 0.364 | 0.364 | 0.364 | 0.364 |

Note: *** indicate statistical significance at the 1% level.

*4.2. Mechanism of Government Attention to the Digital Economy in Influencing DT*

The above empirical evidence shows that the degree of government attention to the digital economy can significantly promote DT. Next, this paper focuses on the mechanisms through which government attention to the digital economy affects DT and analyzes them through the channels of financial technology expenditures, the digital economy level, the digital financial inclusion level, the digital economy industry agglomeration, and the nature of firms.

4.2.1. Channels of Fiscal Science and Technology Expenditure

As a result of government fiscal expenditures on science and technology, there is a compensating effect on the lack of funding for innovation, transformation, and upgrading, thereby reducing the risk faced by enterprises when engaging in innovation. This spending serves to guide companies when investing capital and technology in digital transformation. If innovation activities are handed over to the market, then the innovation investments of enterprises may be insufficient, and the intensity of the innovations may be lower than the ideal societal level. Therefore, government financial spending on science and technology corrects for market failures in innovation activities from the supply side. However, government innovation funding is selective and not available to all firms. Receiving government support implies affirmation from the government and helps firms attract external investments [121]. Fiscal science and technology spending puts into practice the impact of increased government attention on enterprise digital transformation by releasing positive signals. Conversely, firms in regions with high fiscal expenditures pay more attention to government economic, financial, and fiscal policies as they have closer ties with the government and are supported more frequently. Therefore, government attention to the digital economy helps and incentivizes enterprise digital transformation with financial and technological expenditure and, at the same time, releases the signal that the government attaches importance to the digital economy, thereby promoting and reducing the cost of enterprise digital transformation and reducing the number of corporate financing constraints.

This article examines the fiscal science and technology expenditure channel, measured by its spending intensity, and uses two indicators, namely, the ratio of fiscal science and technology expenditure to GDP (STG) and to fiscal revenue (STFR), to ensure the robustness of the results. Data are collected manually from the China Science and Technology Statistical Yearbook and provincial statistical yearbooks based on the information contained in the study.

The interaction terms of GA and STG (STFR) may have a multicollinearity problem, which causes a possible bias in the estimation results. In this paper, GA and government financial expenditures on science and technology are centralized. To facilitate the analysis, the variables analyzed are represented by INTER, as shown in Equation (2):

$$DT = \beta_0 + \beta_1 GA + \beta_2 INTER + \beta_3 AG \times INTER + \gamma \sum Controls + \sum FE + \varepsilon, \quad (2)$$

The following information can be found in Columns (1) and (2) of Table 4: the coefficient of the interaction term between GA and STG is 0.06, which is significant at the 5% level. According to Columns (3) and (4) of Table 4, the coefficient of the interaction term between GA and STFR is 0.01, which is significant at the 1% level. Thus, fiscal science and technology spending significantly increases the extent to which government attention promotes enterprise digital transformation.

**Table 4.** Fiscal science and technology expenditure channels.

|  | (1) | (2) | (3) | (4) |
|---|---|---|---|---|
|  | **STG** | **STG** | **STFR** | **STFR** |
| GA | 0.025 *** | 0.025 ** | 0.028 *** | 0.028 *** |
|  | (2.86) | (2.68) | (3.24) | (3.02) |
| INTER | 0.126 ** | 0.134 ** | 0.007 | 0.007 |
|  | (2.16) | (2.39) | (1.58) | (1.69) |
| GA × INTER | 0.061 ** | 0.062 ** | 0.012 *** | 0.012 *** |
|  | (2.16) | (2.23) | (3.88) | (4.11) |
| HHI | −0.006 | 0.034 | −0.007 | 0.032 |
|  | (−0.32) | (0.79) | (−0.35) | (0.74) |
| FD | 0.006 | 0.007 | 0.022 | 0.024 |
|  | (0.43) | (0.52) | (1.09) | (1.25) |
| ID | 1.028 *** | 1.022 *** | 1.028 *** | 1.024 *** |
|  | (55.23) | (36.95) | (57.22) | (37.84) |
| Size | 0.117 *** | 0.125 *** | 0.117 *** | 0.126 *** |
|  | (8.68) | (9.45) | (8.70) | (9.47) |
| Lev | −0.278 *** | −0.235 *** | −0.278 *** | −0.236 *** |
|  | (−4.54) | (−3.46) | (−4.57) | (−3.46) |
| Growth | 0.112 *** | 0.110 *** | 0.111 *** | 0.110 *** |
|  | (4.88) | (5.03) | (4.91) | (5.04) |
| Board | −0.027 | −0.032 | −0.028 | −0.033 |
|  | (−0.22) | (−0.26) | (−0.22) | (−0.26) |
| Indep | 0.013 | −0.000 | 0.019 | 0.007 |
|  | (0.03) | (−0.00) | (0.05) | (0.02) |
| Dual | 0.105 *** | 0.103 *** | 0.106 *** | 0.104 *** |
|  | (3.69) | (3.55) | (3.68) | (3.55) |
| Constant | −2.441 *** | −2.642 *** | −2.509 *** | −2.720 *** |
|  | (−5.13) | (−4.93) | (−5.42) | (−5.28) |
| Industry FE | No | Yes | No | Yes |
| Year FE | Yes | Yes | Yes | Yes |
| Observations | 13,697 | 13,697 | 13,697 | 13,697 |
| adj. $R^2$ | 0.364 | 0.365 | 0.364 | 0.364 |

Note: *** and ** indicate statistical significance at the 1% and 5% level, respectively.

### 4.2.2. Digital Economy and Digital Financial Inclusion Level Channels

The development level of the digital economy reflects the full combination of a "well-functioning government" and an "effective market". Well-functioning governments can overcome market failures, improve the business environment for enterprises, and build a large number of digital infrastructures in advance, while effective markets can allocate many resources, especially capital, to the digital economy. The digital economy can use information technology to unite information subjects and provide enterprises with good government affairs, rules of law, and market atmosphere, thereby improving enterprise business environments. The digital economy relies on the internet and big data and uses digital technology to help corporate executives and investors capture market information in a timely manner, reduce information asymmetry between banks and enterprises, and alleviate financing constraints caused by insufficient technology and geographical restrictions. As a result of the development of the digital economy and digital financial inclusion, external supervision levels and mass transaction risk control systems can be improved, credit transactions between banks and enterprises can proceed smoothly, and enterprises can be provided with a good guarantee of their digital transformation.

Digital financial inclusion is an important type of financial infrastructure. On the supply side, digital financial inclusion can help enterprises collect and process massive amounts of information, alleviate the misallocation of corporate credit resources, expand diversified financing channels (such as supply chain finance and intelligent investment advisory), and provide enterprises with technical tools to effectively match project risks with their own available resources; hence, such inclusion is conducive to enterprises

making innovative investment decisions. On the demand side, the emergence of mobile payments such as Alipay and WeChat has reconstructed the business model of digital payment platforms, and through the new business ecosystem built by digital technology, enterprises can use technologies such as cloud computing and big data to more accurately depict consumer portraits and effectively capture market demand. Therefore, digital finance has increased the motivation for enterprise digital transformation. At the same time, digital financial inclusion can alleviate the problems of adverse selection and moral hazard among managers by improving the transparency of the corporate environment so that managers' risk-taking ability increases and they pay more attention to new digital technology, thus increasing investment in digital innovation and improving the degree of digital transformation.

Therefore, the higher the levels of the digital economy and digital finance, the richer the basic resources of IT and the higher the availability of funds, providing a good foundation for the digital transformation strategy of enterprises [64].

In this study, we use the Digital Economy Index and Peking University Digital Financial Inclusion Index of China (PKU_DFIIC) to evaluate the level of development of the digital economy and the level of digital financial inclusion [122]. First, this study uses the entropy weight method to calculate the digital economy index (4 internet-development-level indicators and 1 financial inclusion index). Among them, the indicators of internet development include the penetration rate of the internet, the penetration rate of mobile phones, the number of internet practitioners, and related outputs [123]. Next, in this paper, we examine the impact of GA on the Index of the Digital Economy and the Index of Digital Financial Inclusion. Due to data availability, this paper contains analyses of provincial data for 30 provinces, with the exception of Tibet, from 2011 to 2020. The control variables are provincial macro data, which are per capita GDP (lnperGDP = GDP/poplulation), industrial structure (IS = tertiary/secondary industry value added), highway accessibility intensity (HAS = highway length/area of the region), and urbanization rate (UR = urban/total regional population). Since the variables are all at the macrolevel, the fixed effects of province and year are controlled. According to Table 5, government attention to the digital economy is positively correlated with both the Digital Economy Index and the Digital Financial Inclusion Index, and the coefficients are all significant. Therefore, this paper finds that by promoting the development of local digital economies and digital financial inclusion, government attention to the digital economy promotes enterprise digital transformation.

**Table 5.** Digital economy and digital financial inclusion level channels.

| | (1) | (2) | (3) | (4) |
|---|---|---|---|---|
| | Digital Economy Index | Digital Economy Index | Digital Financial Inclusion Index of China | Digital Financial Inclusion Index of China |
| GA | 0.003 * | 0.002 ** | 1.145 ** | 0.799 * |
| | (1.94) | (2.24) | (2.13) | (1.88) |
| lnperGDP | | 0.103 *** | | 40.469 *** |
| | | (10.17) | | (10.37) |
| IS | | 0.028 *** | | −1.603 |
| | | (4.32) | | (−0.63) |
| HAS | | −0.041 *** | | 2.628 |
| | | (−2.84) | | (0.47) |
| UR | | −0.278 *** | | −232.034 *** |
| | | (−3.91) | | (−8.45) |
| Constant | 0.270 *** | −0.628 *** | 101.977 *** | −163.136 *** |
| | (39.01) | (−5.26) | (36.56) | (−3.53) |

**Table 5.** *Cont.*

|  | (1) | (2) | (3) | (4) |
|---|---|---|---|---|
|  | **Digital Economy Index** | **Digital Economy Index** | **Digital Financial Inclusion Index of China** | **Digital Financial Inclusion Index of China** |
| prov | Yes | Yes | Yes | Yes |
| year | Yes | Yes | Yes | Yes |
| N | 300 | 300 | 300 | 300 |
| adj. $R^2$ | 0.990 | 0.993 | 0.995 | 0.997 |

Note: ***, **, and * indicate statistical significance at the 1%, 5%, and 10% level, respectively.

### 4.2.3. Digital Economy Industry Aggregation Channel

When the government attaches importance to digital transformation, it builds digital economy industrial parks on a large scale, attracts investments, promotes economic growth, and provides preferential policies, such as preferential financing, preferential taxation, and preferential land policies, to local firms to create conditions for the industrial agglomeration of firms and capital flows toward firms for their digital transformation. Regions with a high degree of industrial agglomeration in the digital economy promote enterprise digital transformation through knowledge spillover, catch-up pressure, and external economies of scale. External economies of scale reduce the costs of collecting raw materials and arising from transactions, lower firms' production costs, make it easier to obtain stable supplier services, and enable collaborative transformation upstream and downstream of the industry chain. External economies of scale can also promote divisions of labor and cooperation, improve the efficiency of collaboration, enhance labor productivity, and reduce employment costs for firms. Geographical agglomeration in industrial parks also reduces the cost of government policy implementation. Knowledge technology spillover and knowledge sharing can encourage firms to adopt new digital technologies, allowing them to communicate more easily and interact more frequently, thus allowing new technologies to spread faster. Firms in the same region engaging in comparisons generate catch-up pressure, encouraging them to reduce their costs to improve product services and catch up with technological hotspots.

In this paper, we measure the industrial agglomeration channel and study the impact of industrial agglomeration on the digital economy. The digital economy industry includes "computer, communication, and other electronic equipment manufacturing", which is digital industrialization, and "information transmission, software, and information technology services", which is the digitalization part of the industry in China. Due to data availability, this paper presents data on the industrial clustering of firms in the core industry of the digital economy (digital industrialization) from 2011 to 2020. This subsection collates the data of 30 provinces, except Tibet, and counts the number of employees (10,000) in the manufacturing of communication equipment, computers, and other electronic equipment related to the digital economy [124,125]. The data sources are provincial statistical yearbooks or employment statistical yearbooks and are compiled manually. The digital economy industry agglomeration ($S$) is measured using the locational entropy method. Since there are no data on the value added to industry by subsector in the statistical yearbooks, $S$ is calculated by the employed population, $S_i = (E_{i,m}/E_i)/(E_m/E)$, where $E_{i,m}$ represents overall employment in digital economy-related industries in province $i$, $E_i$ represents overall employment in province $i$, $E_m$ represents overall employment in digital economy-related industries nationwide, and $E$ is total national employment in China.

Subsequently, industrial agglomeration S is grouped according to the mean. As seen from Table 6, in Columns (1) and (2), in areas with high industrial agglomeration in the digital economy, the greater the government's attention to the digital economy is, the greater its impact on enterprise digital transformation. Therefore, GA has a significant impact only on channels with high industrial agglomeration.

**Table 6.** Digital economy industrial clustering and firm nature channels.

|  | (1) | (2) | (3) | (4) |
|---|---|---|---|---|
|  | High Industrial Agglomeration | Low Industrial Agglomeration | SOEs | Non-SOEs |
| GA | 0.058 * | 0.008 | 0.011 | 0.032 *** |
|  | (1.85) | (0.71) | (0.63) | (4.16) |
| HHI | 0.132 | −0.016 | −0.036 | 0.105 |
|  | (0.91) | (−0.16) | (−0.29) | (0.90) |
| FD | 0.038 * | 0.015 | −0.013 | 0.047 ** |
|  | (2.04) | (0.69) | (−0.61) | (2.47) |
| ID | 0.895 *** | 1.093 *** | 1.084 *** | 0.992 *** |
|  | (16.86) | (40.01) | (16.86) | (23.05) |
| Size | 0.145 *** | 0.117 *** | 0.111 *** | 0.160 *** |
|  | (7.07) | (6.25) | (4.61) | (6.70) |
| Lev | −0.130 | −0.296 *** | −0.364 *** | −0.174 |
|  | (−0.86) | (−4.07) | (−4.00) | (−1.63) |
| Growth | 0.059 * | 0.135 *** | 0.085 | 0.090 *** |
|  | (1.74) | (4.54) | (1.12) | (4.94) |
| Board | −0.123 | 0.014 | −0.052 | 0.047 |
|  | (−1.30) | (0.10) | (−0.30) | (0.29) |
| Indep | −0.130 | 0.007 | 0.511 | 0.093 |
|  | (−0.29) | (0.01) | (0.86) | (0.24) |
| Dual | 0.068 | 0.131 * | 0.139 | 0.064 * |
|  | (1.31) | (1.97) | (1.70) | (1.98) |
| Constant | −2.972 *** | −2.705 *** | −2.244 *** | −3.942 *** |
|  | (−4.61) | (−4.84) | (−6.08) | (−4.34) |
| Industry FE | Yes | Yes | Yes | Yes |
| Year FE | Yes | Yes | Yes | Yes |
| Observations | 5336 | 8361 | 3775 | 9922 |
| adj. $R^2$ | 0.303 | 0.404 | 0.359 | 0.361 |

Note: ***, **, and * indicate statistical significance at the 1%, 5%, and 10% level, respectively.

### 4.2.4. Firm Nature Channel

China's state-owned enterprises (SOEs) have advantages in the industrial chain: more sufficient profits, more abundant capital, lower financing constraints, and less competitive pressure than non-state-owned enterprises. Therefore, state-owned enterprises are more lenient in screening investment projects and do not necessarily choose digital transformation projects with better prospects. At the same time, SOEs have closer ties with the government and may invest in other inefficient projects to obtain government funding support and thus cater to the government, which has a crowding-out effect on digital transformation projects. SOEs are not as flexible in their management approach as non-SOEs and have insufficient demand and motivation for transformation and upgrading. Non-SOEs have a strong need to improve profits and transformations and are more willing to cooperate with government policies related to digital transformation. Therefore, the role of GA in promoting digital transformation is more significant for non-state-owned enterprises. In this paper, we measure the firm nature channel and analyze the effect of whether the firm is a state-owned enterprise (SOE). The SOE value of the firm is one if it is state-owned and zero otherwise. As seen in Columns (3) and (4) in Table 6, at the 1% significance level, GA to non-state-owned enterprises has a positive effect on DT, while that of state-owned enterprises is not significant; thus, government attention to the digital economy of non-state-owned enterprises has a more obvious impact on DT.

### 4.2.5. Further Analysis: Market Competition and Enterprise Size

From the benchmark regression, the main factor driving DT is government attention. We divide firms into two groups according to the median size and then re-analyze Formula (1). Columns (1) and (2) in Table 7 are regressions for a sample of large firms, and

Columns (3) and (4) are for small firms. Further grouping of firms according to their median size shows that government attention to the digital economy has a positive effect on DT for large firms, and this is significant at the 1% level, whereas there is no significant effect for small firms. As can also be seen from Columns (3) and (4), the impact of market competition on DT is significantly positive among small enterprises. However, the coefficient for small businesses is not significant and is consistent with benchmark regression. The results show that the role of market competition in DT is influenced mainly by size. As seen from Table 7, DT is positively influenced by the degree of competition among small firms. Government attention affects primarily the DT of large firms, whereas market competition affects primarily the DT of small firms. The possible reason for this is that large firms have stable markets and financing and are motivated to transform and upgrade only under the influence of government attention and preferential policies. Compared with small enterprises, large enterprises can better convert external resources into unique capabilities, which is conducive to enterprise digital transformation [64]. However, small-scale enterprises with weak competitive positions face greater survival problems, a higher degree of information asymmetry, and a weaker voice in sales and procurement [126]. These problems have led to small enterprises needing to pursue DT only under the influence of a higher degree of competition. Overall, the above finding verifies Hypothesis 2 of this paper.

**Table 7.** Large- and small-scale firms.

| | (1) | (2) | (3) | (4) |
|---|---|---|---|---|
| | Large-Scale Firms | Large-Scale Firms | Small-Scale Firms | Small-Scale Firms |
| GA | 0.068 *** | 0.039 *** | 0.016 | 0.015 |
| | (4.39) | (3.21) | (0.99) | (1.08) |
| HHI | 0.067 | −0.104 | 0.263 * | 0.292 ** |
| | (0.58) | (−1.20) | (1.98) | (2.83) |
| FD | | 0.018 | | 0.023 |
| | | (0.66) | | (1.25) |
| ID | | 1.072 *** | | 0.978 *** |
| | | (31.56) | | (19.31) |
| Size | | 0.089 *** | | 0.243 *** |
| | | (5.21) | | (5.23) |
| Lev | | −0.274 ** | | −0.171 |
| | | (−2.62) | | (−1.29) |
| Growth | | 0.090 ** | | 0.104 *** |
| | | (2.60) | | (3.13) |
| Board | | −0.043 | | 0.011 |
| | | (−0.41) | | (0.04) |
| Indep | | −0.209 | | 0.573 |
| | | (−0.39) | | (1.17) |
| Dual | | 0.180 *** | | 0.048 * |
| | | (3.01) | | (1.88) |
| Constant | 0.236 | −2.050 *** | 0.264 | −5.443 *** |
| | (1.51) | (−3.63) | (1.50) | (−4.02) |
| Industry | Yes | Yes | Yes | Yes |
| Year | Yes | Yes | Yes | Yes |
| N | 7642 | 7380 | 7514 | 6571 |
| Adj. $R^2$ | 0.255 | 0.380 | 0.263 | 0.356 |

Note: ***, **, and * indicate statistical significance at the 1%, 5%, and 10% level, respectively.

A comparison of high-tech and non-high-tech firms, as shown in Table 8, indicates that government attention significantly influences enterprise digital transformation in non-high-tech firms. The coefficient of GA in Column (2) is positive and significant, indicating that non-high-tech firms are more likely to be led by the government to improve enterprise digital transformation. In Column (3), there is no significant effect of government attention to the digital economy on enterprise digital transformation, probably because enterprises

in the digital economy are already undergoing digital transformation and are not affected much by the government. Conversely, the coefficient of GA in a non-digital economy is shown to be significant at the 1% level in Column (4). In this regard, it can be concluded that firms that are not part of the digital economy are more likely to be affected by the government's digital economy policies and seek to transform and upgrade their businesses.

**Table 8.** High-tech and digital economy firms.

| | (1) | (2) | (3) | (4) |
|---|---|---|---|---|
| | High-Tech Enterprises | Non-High-Tech Enterprises | Digital Economy Enterprises | Non-Digital Economy Enterprises |
| GA | 0.029 | 0.036 ** | 0.014 | 0.038 *** |
| | (0.70) | (2.70) | (1.14) | (3.54) |
| HHI | 0.399 | −0.032 | 0.326 *** | −0.008 |
| | (1.06) | (−0.51) | (28.22) | (−0.11) |
| FD | 0.042 | 0.019 | 0.064 | −0.005 |
| | (0.92) | (1.05) | (1.43) | (−0.32) |
| ID | 0.889 *** | 1.089 *** | 1.194 *** | 1.039 *** |
| | (14.94) | (69.37) | (11.98) | (76.65) |
| Size | 0.164 *** | 0.107 *** | 0.200 *** | 0.104 *** |
| | (4.50) | (9.52) | (7.01) | (9.85) |
| Lev | −0.209 * | −0.273 *** | −0.194 *** | −0.261 ** |
| | (−1.95) | (−2.93) | (−12.07) | (−2.48) |
| Growth | 0.127 ** | 0.105 *** | 0.065 | 0.118 *** |
| | (2.59) | (4.71) | (0.99) | (6.85) |
| Board | −0.256 | 0.016 | −0.144 | 0.012 |
| | (−1.37) | (0.14) | (−1.17) | (0.08) |
| Indep | −1.024 | 0.364 | 0.121 | 0.019 |
| | (−1.14) | (0.91) | (0.52) | (0.03) |
| Dual | 0.140 *** | 0.106 ** | 0.078 * | 0.120 *** |
| | (3.27) | (2.11) | (2.51) | (3.55) |
| Constant | −2.951 ** | −2.672 *** | −4.665 ** | −2.342 *** |
| | (−2.36) | (−6.05) | (−5.03) | (−3.93) |
| Industry | Yes | Yes | Yes | Yes |
| Year | Yes | Yes | Yes | Yes |
| N | 3439 | 10,258 | 3630 | 10,067 |
| Adj. $R^2$ | 0.351 | 0.373 | 0.296 | 0.214 |

Note: ***, **, and * indicate statistical significance at the 1%, 5%, and 10% level, respectively.

In Table 9, firms are categorized according to their lifecycle. There is evidence that government attention to the digital economy has a significant positive impact on DT during periods of growth and recession. However, there is no significant influence during mature periods, indicating that firms adopt different approaches during different lifecycle stages. Growth-period firms are developing rapidly, have a better operating environment, and are more inclined to carry out digital transformation and improve profits through government attention than firms in other periods. Firms in recession periods have reduced their business capacity, and thus, it has become necessary for them to improve their performance through digital transformation, on which government attention has a significant impact. While mature enterprises are developing steadily, their transformation needs are not as high as in the growth and recession periods, and thus, the government's attention to the digital economy in their case has a limited impact.

**Table 9.** Business lifecycles.

|  | (1) | (2) | (3) |
|---|---|---|---|
|  | **Growth Period** | **Mature Period** | **Recession Period** |
| GA | 0.038 ** | 0.027 | 0.028 * |
|  | (2.58) | (1.69) | (1.78) |
| HHI | 0.129 | −0.242 | 0.214 |
|  | (1.18) | (−1.36) | (1.13) |
| FD | 0.041 ** | 0.009 | 0.005 |
|  | (2.34) | (0.49) | (0.16) |
| ID | 0.992 *** | 1.033 *** | 1.074 *** |
|  | (74.31) | (26.33) | (20.90) |
| Size | 0.103 *** | 0.139 *** | 0.155 *** |
|  | (8.12) | (9.76) | (5.52) |
| Lev | −0.242 | −0.325 *** | −0.251 ** |
|  | (−1.53) | (−3.79) | (−2.33) |
| Growth | 0.091 ** | 0.085 | 0.084 |
|  | (2.68) | (1.45) | (1.45) |
| Board | −0.046 | −0.029 | −0.001 |
|  | (−0.40) | (−0.16) | (−0.01) |
| Indep | −0.064 | 0.006 | 0.159 |
|  | (−0.11) | (0.02) | (0.30) |
| Dual | 0.128 *** | 0.084 ** | 0.090 ** |
|  | (3.55) | (2.44) | (2.19) |
| Constant | −2.370 *** | −2.973 *** | −3.374 *** |
|  | (−5.68) | (−6.31) | (−4.67) |
| Industry | Yes | Yes | Yes |
| Year | Yes | Yes | Yes |
| N | 6188 | 4877 | 2827 |
| Adj. $R^2$ | 0.360 | 0.381 | 0.349 |

Note: ***, **, and * indicate statistical significance at the 1%, 5%, and 10% level, respectively.

### 4.3. Analyses of Robustness

#### 4.3.1. Adjusting the Sample Period

Since our sample period is 2011–2020, to make the results more robust, we shorten the sample period and select 2012–2020 and 2013–2020 data for robustness testing.

As seen in Table 10, government attention to the digital economy still has a significant positive effect on enterprise digital transformation, and Columns (1) to (4) show that industry competitiveness has a non-significant effect on digital transformation. The coefficients for Columns (1) to (4) range from 0.030 to 0.032 and are all significant at the 5% level. This study's findings are robust, as they are consistent with those of the baseline regression.

**Table 10.** Adjustment to the sample period.

|  | (1) | (2) | (3) | (4) |
|---|---|---|---|---|
|  | **2012** | **2012** | **2013** | **2013** |
| GA | 0.030 ** | 0.030 ** | 0.031 ** | 0.032 ** |
|  | (2.34) | (2.73) | (2.33) | (2.63) |
| HHI | 0.039 | −0.001 | 0.045 | 0.009 |
|  | (0.62) | (−0.03) | (0.74) | (0.24) |
| FD | 0.002 | −0.000 | 0.002 | −0.001 |
|  | (0.13) | (−0.01) | (0.15) | (−0.04) |
| ID | 1.068 *** | 1.046 *** | 1.063 *** | 1.045 *** |
|  | (49.75) | (111.56) | (51.82) | (104.37) |
| Size | 0.123 *** | 0.111 *** | 0.124 *** | 0.110 *** |
|  | (5.28) | (5.50) | (5.13) | (4.92) |
| Lev | −0.165 ** | −0.241 *** | −0.153 * | −0.219 ** |
|  | (−2.26) | (−3.23) | (−2.02) | (−2.74) |

**Table 10.** *Cont*.

|  | **(1)** | **(2)** | **(3)** | **(4)** |
|---|---|---|---|---|
|  | **2012** | **2012** | **2013** | **2013** |
| Growth | 0.111 *** | 0.112 *** | 0.106 *** | 0.109 *** |
|  | (3.86) | (3.81) | (3.66) | (3.72) |
| Board | 0.091 | 0.080 | 0.094 | 0.099 |
|  | (0.75) | (0.71) | (0.76) | (0.84) |
| Indep | 0.237 | 0.227 | 0.291 | 0.318 |
|  | (0.41) | (0.40) | (0.48) | (0.52) |
| Dual | 0.083 * | 0.083 ** | 0.078 * | 0.083 * |
|  | (2.03) | (2.28) | (1.93) | (2.05) |
| Constant | −3.056 *** | −2.752 *** | −3.158 *** | −2.829 *** |
|  | (−5.05) | (−4.59) | (−4.95) | (−4.58) |
| Industry FE | Yes | No | Yes | No |
| Year FE | Yes | Yes | Yes | Yes |
| Observations | 8929 | 9324 | 8629 | 8629 |
| Adj. $R^2$ | 0.441 | 0.437 | 0.440 | 0.440 |

Note: ***, **, and * indicate statistical significance at the 1%, 5%, and 10% level, respectively.

### 4.3.2. Independent Variables Lagged by One and Two Periods

GA lagged by one and two periods are used as independent variables for regression, the results of which are presented in Table 11. There is a positive correlation between GA coefficients in the first to fourth columns, and all the correlation coefficients are significant at the 5% or 1% level. The results remain robust with lagged independent variables.

**Table 11.** Independent variables lagged by one and two periods.

|  | **(1)** | **(2)** | **(3)** | **(4)** |
|---|---|---|---|---|
|  | **One-Period Lag** | **One-Period Lag** | **Two-Period Lag** | **Two-Period Lag** |
| GA | 0.032 ** | 0.032 ** | 0.029 ** | 0.030 *** |
|  | (2.13) | (2.35) | (2.67) | (2.94) |
| HHI | 0.027 | 0.005 | 0.031 | 0.026 |
|  | (0.43) | (0.14) | (0.83) | (0.67) |
| FD | 0.002 | −0.002 | −0.000 | −0.003 |
|  | (0.12) | (−0.11) | (−0.02) | (−0.20) |
| ID | 1.067 *** | 1.048 *** | 1.071 *** | 1.050 *** |
|  | (44.15) | (95.82) | (36.81) | (84.26) |
| Size | 0.124 *** | 0.112 *** | 0.123 *** | 0.109 *** |
|  | (5.26) | (5.04) | (5.29) | (5.00) |
| Lev | −0.173 ** | −0.245 *** | −0.227 ** | −0.295 *** |
|  | (−2.31) | (−3.16) | (−2.78) | (−3.47) |
| Growth | 0.110 *** | 0.114 *** | 0.089 ** | 0.094 ** |
|  | (3.26) | (3.35) | (2.51) | (2.65) |
| Board | 0.084 | 0.091 | 0.091 | 0.098 |
|  | (0.66) | (0.74) | (0.69) | (0.79) |
| Indep | 0.236 | 0.261 | 0.351 | 0.382 |
|  | (0.41) | (0.45) | (0.54) | (0.58) |
| Dual | 0.091 ** | 0.094 ** | 0.089 * | 0.092 * |
|  | (2.14) | (2.21) | (1.93) | (2.00) |
| Constant | −3.089 *** | −2.766 *** | −3.124 *** | −2.772 *** |
|  | (−5.31) | (−4.88) | (−5.12) | (−4.53) |
| Industry | Yes | No | Yes | No |
| Year | Yes | Yes | Yes | Yes |
| N | 8669 | 8669 | 7910 | 7910 |
| Adj. $R^2$ | 0.442 | 0.442 | 0.443 | 0.443 |

Note: ***, **, and * indicate statistical significance at the 1%, 5%, and 10% level, respectively.

## 5. Conclusions

Based on press release data, this paper studies the relationships among government attention to the digital economy, market competition, and enterprise digital transformation. The empirical analysis results show that, first, government attention to the digital economy has a significant impact on enterprise digital transformation and that government attention significantly promotes enterprise digital transformation in the region, while the role of competition is not significant for the whole sample. The above conclusions still hold after robustness analysis, which involves reducing the analysis window and replacing the independent variables. Second, the group analysis finds that only in small-scale groups can competition have a significant positive impact on DT. It is evident from this result that the role of competition in driving digital transformation depends on the size of the enterprise. Third, regarding the mechanism, fiscal science and technology expenditures and a high degree of digital economy industry agglomeration can significantly promote the degree of government attention affecting enterprise digital transformation, and GA promotes enterprise digital transformation by promoting the local digital economy level and digital financial inclusion level. GA has a significant impact on digital transformation only for non-state-owned enterprises. Fourth, the heterogeneity analysis shows that government attention has a significant impact on DT in non-high-tech enterprises, non-digital economy enterprises, growth enterprises, and recession enterprises.

This study makes the following policy recommendations: First, government attention to the digital economy should be increased. The government needs to set up a professional digital economy team, coordinate various departments, promote data sharing and openness, and establish a long-term communication mechanism between itself and enterprises. It also needs to provide professional digital publicity and training services to society. In addition, the government should deepen its digital reform, improve its efficiency, and transform into an efficient, public service-oriented government. Second, the digital transformation infrastructure should be improved. The government should promote digital industry agglomeration, establish digital economy industrial parks, and develop a data sharing platform. Moreover, the government should encourage industry-university-research collaboration among the government, enterprises, and universities and then introduce high-quality digital talent to provide assurance for enterprise digital transformation. Third, more fiscal and tax policies related to the digital economy should be introduced. The government should increase fiscal spending on science and technology; provide credit, taxes, and incentives; improve the financial status of enterprises; and provide sufficient financial guarantees for enterprise digital transformation. The fourth recommendation is that the supervision of the digital economy and digital financial inclusion by the government be improved. The government should issue laws and regulations to address possible data leakage problems, establish a sound data trading market, and ensure fair competition in the market. Fifth, government attention should be paid to the digital transformation and upgrading of non-state-owned enterprises and non-digital economy firms.

There are some potential shortcomings in this article. First, in terms of objective factors, the sample range of this paper is from 2011 to 2020, and future studies can broaden the sample period to obtain richer conclusions. Second, this paper studies data at the provincial level. In the future, studies can be conducted at the city or county level to draw more detailed conclusions. Third, in terms of mechanism analysis, this paper preliminarily analyzes the factors driving enterprise digital transformation, but there is no empirical analysis of the direct impact of government attention on the digital economy with respect to local digital economy infrastructure, enterprise investments and financing, corporate governance, etc. These aspects can be broadened in the future. Finally, the government's conclusion that attention to the digital economy affects enterprise digital transformation is consistent with China's development of the digital economy. However, this situation may not be applicable in other countries. In future studies, other intervention variables should be included based on different application environments.

**Author Contributions:** Conceptualization, X.J.; Methodology, X.J.; Software, X.P.; Validation, X.P.; Formal analysis, X.P.; Investigation, X.P.; Resources, X.J.; Data curation, X.P.; Writing—original draft, X.P.; Writing—review & editing, X.J.; Visualization, X.J.; Supervision, X.J.; Project administration, X.P. All authors have read and agreed to the published version of the manuscript.

**Funding:** This research received no external funding.

**Institutional Review Board Statement:** Not applicable.

**Informed Consent Statement:** Not applicable.

**Data Availability Statement:** The dataset generated and analyzed in this study is not publicly available. Dataset is available from the corresponding author on reasonable request.

**Conflicts of Interest:** The authors declare no conflict of interest.

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
