# Peer review of "Government Attention, Market Competition and Firm Digital Transformation"

_sustainability, doi:10.3390/su15119057_

Round 1

Reviewer 1 Report

Dear Authors,

Congratulations on a very successful development. The text is very interesting, it describes the issues related to digital transformation in an innovative way. Noteworthy is the interesting methodology and original scenario of the research procedure.

Please complete one strand in the text: I have not found in the text a clearly formulated comment to hypothesis number 2. Please fill in this lack.

Reviewer 2 Report

China is a country of large numbers, so research from this market is very valuable to us. This is also reflected in this paper, where the sample is based on "38,891 news items from provincial and municipal governments in China".

The speed of emergence and the degree of revolutionary scientific and technological inventions coming from research centers, startups companies and small scale or large organizations are worldwide. New opportunities that dynamic technological progress provides also changes the way we live and work. This also affects the way we manage our business. The rules of business have changed.

Theoretical issues and research hypotheses in correlation with research design are well established.

The authors raise the question of how to choose the right digital transformation strategy and whether the right measure has been achieved in the role of the state. To which they provide answers about the current situation on the studied part of the market.

The paper has value and makes a contribution. However, there is still room for improvement, and I give the authors few suggestions:

  • How long have the authors been doing this research?
  • Line 108 - 111 insert a map of China and mark the 31 provinces where the research was done. In this way, the size of the research territory is more closely explained to the readers.
  • Digital transformation and technological innovation have led to new professions that require new skills and knowledge in various industries. Briefly refer to the case in China.

The paper provides recommendations that should form the backbone of the author's future papers.

Since the authors researched the sample range from 2011 to 2020, it is necessary to refer very briefly to the impact of Covid19 on the digitization process, if applicable. If not, I suggest the authors to do new research in next paper for the period 2020-2023 and show this impact.

Reviewer 3 Report

Dear Authors

The authors of the article examine the impact of government policy on the digital transformation of enterprises. They point out that digital transformation is an important part of economic development. They indicate tools (e.g. fiscal, tax) that directly support digital transformation. The literature review is quite extensive, but it would be worth adding more studies on the digital transformation of European and American authors. The empirical part is prepared correctly. The authors put forward two research hypotheses. They correctly define the dependent, independent and control variables. However, the results are quite obvious. They do not surprise with anything new, but the method of selecting variables and data as well as the method of proving deserves a positive assessment. The authors also identify the shortcomings of the article well, pointing to directions for further research. Perhaps it is also worth paying attention to the aspect of using digital technologies for the circular economy and, more broadly, for environmental protection. As a consequence, this may also lead to a positive impact on the development of enterprises and additionally protect the natural environment. An important aspect of digitization also concerns human resource management, which undoubtedly also affects the development of enterprises. Please pay attention to such articles as: https://doi.org/10.1016/j.joitmc.2023.01.001 ; https://sciendo.com/fr/article/10.2478/mspe-2022-0023 ; https://doi.org/10.3390/en15010172

Good luck with your further research

Reviewer

Reviewer 4 Report

This articel presents a good analysis based on weak premises. The main problem is the unsufficient review of the litereature. The theoretical framework is vauge, so to undermine the research question and the hypotheses developmet. This study needs many more references that define the problem and the concepts.

The argument is inadeqaute. Two ways: “promotion of market competition and the promotion of government attention” (row 40): who said that? “the invisible hand of the market” (row 41), this must be updated, and avoid this kind of triviality. Proved wrong and misleading. Firms compete, mimic each other, and influence the market. Very superficial way of introducing the problem. Morover, “industrial digitization and digital industrialization” what is exactly the difference? Do you mean  industrialization  of the digital economy/technology?

Everything between line 49 and 107 regarde literature review, and should  stay in the next section. 

Hard to agree on  “The literature on firms’ digital transformation focuses on the impact of such transformations on firms and macroeconomics or studies the impact of being governed by a digital government on society and the economy, given the digitization of the government itself; however, analysis from the perspective of the transformation of the drivers of firms’ digital transformation drivers is lacking, and only a few papers have delved into the segmentation of financial technology expenditures and government subsidies.” (115-120) The authors do not cite any works, and worst, do not prove the existence of a gap in the literature.

What is the defition of government attention in the literature?

“When a local government pays attention to a certain aspect, it investigates and studies this aspect, promulgates a series of plans and policies, and accelerates the approval of government-led projects in the field.” It does not convince the reader.

An example in the literature is Herbert Simon definition of attention as "the process in which managers selectively pay attention to some information while ignoring other parts" (Simon, 1976). With reference to the government, a definition is “"attention-driven policy choice model" Jones et al. (1993)

Hypothesis 1. Digital transformation of enterprises is significantly facilitated by increased government attention to the digital economy.

Following your framework, and more logicallt, H1 should be: “government attention to the digital economy eases the digital transforation of enterprises.” Avoid passive form

References does not look adequate. Example, [57] for “The willingness to adopt new technologies” …

Simon, H. A. (1976). Administrative behavior: A study of decision-making processes in administrative organization

Jones, B. D., Baumgartner, F. R., & Talbert, J. C. (1993). The destruction of issue monopolies in Congress. American Political Science Review87(3), 657-671.

This manuscript needs a revision of the language. The use of the passive voice does not help the comprehension.

Round 2

Reviewer 4 Report

I must praise the authors for addressing all the remarks I raised in the first review of this paper. Despite the improvement, however, I still have a few comments that I hope could help the authors in refreshing their work.

First, a great part of what is in the introduction belongs to other sections. It is unusual to see figures and maps in the introductory section. I suggest moving the map and figure together with the descriptions to the section on methods and data.

Second, while I appreciate the expansion in the references cited, I still think that section 2 is not a "literature review". Refers to it as just the "theoretical framework" for your research. 

All the best.

A great improvement in style compared to the first version.
